# Economic Valuation of Mangroves and a Linear Mixed Model-Assisted Framework for Identifying Its Main Drivers: A Case Study in Benin

**Corine Bitossessi Laurenda Sinsin** [1,2] , **Alice Bonou** [3,*] , **Kolawolé Valère Salako** [1] , **Rodrigue Castro Gbedomon** [1,4] **and Romain Lucas Glèlè Kakaï** [1]

1 Laboratoire de Biomathématiques et d'Estimations Forestières, Faculté des Sciences Agronomiques, Université d'Abomey-Calavi, Cotonou 04 BP 1525, Benin; corine.sinsin@leibniz-zmt.de (C.B.L.S.); valere.salako@fsa.uac.bj (K.V.S.); rodrigue.gbedomon@unige.ch (R.C.G.); romain.glelekakai@fsa.uac.bj (R.L.G.K.)
2 Leibniz Centre for Tropical Marine Research (ZMT), Fahrenheitstraße 6, 28359 Bremen, Germany
3 École d'Agrobusiness et de Politiques Agricoles, Université Nationale d'Agriculture, Porto-Novo 01 BP 55, Benin
4 Institute for Environmental Sciences, University of Geneva, 66 Boulevard Carl-Vogt, 1205 Geneva, Switzerland
* Correspondence: alice.bonou@una.bj

**Abstract:** Mangroves are brackish wetland ecosystems found in tropical areas. They are highly productive ecosystems that contribute to the economic empowerment of local communities. Proper estimation of their monetary value and the extent of their contribution to rural households' income, although challenging, is paramount for sustainable management decisions. This study aimed to estimate the total economic wealth earned from mangrove ecosystems in Benin. Specifically, the study assessed the diversity of ecosystem services (ESs) provided by mangroves and the contribution of ESs to the total annual economic value of mangroves, and it identified socio-demographic drivers of the total economic value at the individual informant level. In total, 298 informants from 15 villages were interviewed to determine the diversity of mangrove ESs. The ESs were then gathered per category. Household-level economic values of mangroves, economic values of mangroves per ES category, and total economic value were estimated by combining diverse approaches. The contribution of each category of ES to the total economic value (TEV) was determined. A Principal Component Analysis (PCA) was applied to describe the relationships between the economic value of categories of ESs. A Linear Mixed Effect Model (LMEM) was used to determine valid socio-demographic drivers of the TEV. Twenty-nine ESs were identified, with regulation and recreation services being the best contributors to annual TEV, which was estimated at USD 1.29 billion (USD 195,223.69/hectare). Stakeholdership followed by household size are the main socio-demographic drivers of TEV. The identified ESs and their estimated economic value can be incorporated into policy briefs and technical sheets to (i) promote ESs for the optimisation of TEV and (ii) raise awareness and funding for the conservation and sustainable management of mangrove ecosystems.

**Keywords:** mangroves; ecosystem services; economic value; climate negotiations; statistical models





## 1. Introduction

Mangroves are brackish wetland ecosystems found in the tropics. Like coral reefs, they are recognised as one of the most productive ecosystems of the world [1,2]. Indeed, their productivity merits are revealed by yields from ecosystem services (ESs) such as fisheries (fish, crabs, oysters, shrimp, etc.), timber and derivatives, fresh air, clean and oxygenised water, shoreline protection, etc. [3–5]. Mangroves contribute to poverty alleviation [6], food security [7], rural woman empowerment, climate regulation, and ecosystem-based adaptation to climate change [8]. Mangroves are among the most threatened ecosystems [9]. Threats to mangroves result from the low interest seen in conservation policies

that may have favoured intensive logging, overfishing, and unsustainable management paths. Quantification of the value of ecosystems is often used in association with other management tools to orient policy decisions and management plans. For example, the economic importance of mangroves from Asia has widely been investigated, and outcomes of such investigations were used to account for mangroves in conservation and economic development programs [10,11]. For example, [12], [13], and [14], respectively, assessed the economic values of mangroves in South Asia, Vanuatu, and Can Gio. Their estimations have been used to improve the conservation merits of mangroves in these areas. These examples show that the economic valuation of mangroves can support policy decisions and conservation debates.

West Africa is also home to significant mangroves concentrated in countries such as Nigeria, The Gambia, Senegal, Ghana, Côte d'Ivoire, Benin, Liberia, Sierra Leonne, Guinea, Guinea Bissau, and Togo. The existing economic valuation of West African mangroves is restricted to countries such as The Gambia [15], Nigeria [6], and Ghana [16]. There is no comprehensive and detailed report on West African mangroves' economic valuation. One way to do so is to provide a clear and thorough assessment of each country where such information is unavailable. To our knowledge, such information is lacking in many countries in the region, including Benin.

The wide variability of methods used to estimate the monetary value of mangrove ESs belongs to four categories, namely, (i) market-based valuation, (ii) revealed preference methods, (iii) stated preference methods, and (iv) benefits transfer [17]. However, there is no ready-to-take method, and the choice of methodology should be carefully made depending on the nature of the ES [18,19]. Thus, any consistent economic valuation of an ecosystem requires prospective investigations using a checklist for all ESs procured [20,21]. Depending on the nature of recorded ESs—direct use, indirect use, non-use, optional, existence, and bequest—the appropriate specific method is used for the economic valuation. Market price, substitute price, travel cost, and contingent valuation are standard techniques used to estimate the price of goods and services from mangroves [18,22]. Substitute price cost and contingent valuation often overestimate the value of services because of their non-market character [22]. However, mangroves' structural and functional complexities have led to the overuse of the benefit transfer method and less attention to cultural ecosystem services that are often specific to human communities living in the surroundings [23]. The benefits transfer method (meta-regression) may overlook the social realities of the study area [24]. For example, [18] reported ground data deficiency and inconsistencies in global economic values for mangroves.

Benin is a West African country where mangroves cover about 6600 ha, and the ecosystem is dominated by species such as *Rhizophora racemosa* and *Avicennia germinans* [25]. Information on ESs provided by the ecosystem and their economic value is expected to assist efforts toward its conservation and sustainable management. This study aimed to assess the total economic value of mangroves in Benin, combining several approaches and accounting for all ESs provided by the ecosystem. Specifically, the study (i) inventoried ecosystem services procured by mangroves in Benin, (ii) estimated the monetary value of each category of services and the global annual wealth earned from mangroves in Benin, and (iii) identified the main drivers of the mangroves' economic value at an informant level.

## 2. Material and Methods

### 2.1. Study Area

The study was conducted in all mangrove sites of Benin, which were along lakes and lagoons (Figure 1). The study sites consist of a habitat dominated by *Rhizophora racemosa* (G.) Meyer, followed by *Avicennia germinans* (L.) Leechm. Other mangrove species include *Conocarpus erectus* (L.), *Laguncularia racemosa* (L.) C.F. Gaertn, and *Acrostichum aureum* L. These harbour exceptionally high biodiversity with rich wetland flora of 364 species belonging to 100 families [16] and fauna of marine and inland species, including birds and reptiles, small mammals, rodents, etc. [16]. Human communities include many

ethnic groups: the Goun and Tori, around Aguégué and Porto Novo; the Pla, between Pahou and Avlékété; the Pédah, between Ouidah and Djègbadji; the Aizo, between Cococodji and Godomey; and the Fon and the Keta found along the coast and elsewhere [16]. Maritime fishing and fishing in nearby lakes and lagoon systems are the most prominent human activities. Mangrove wood collection for firewood, salt production, charcoal production, and construction is an everyday activity [16]. Some local conservation NGOs also promote touristic activities.

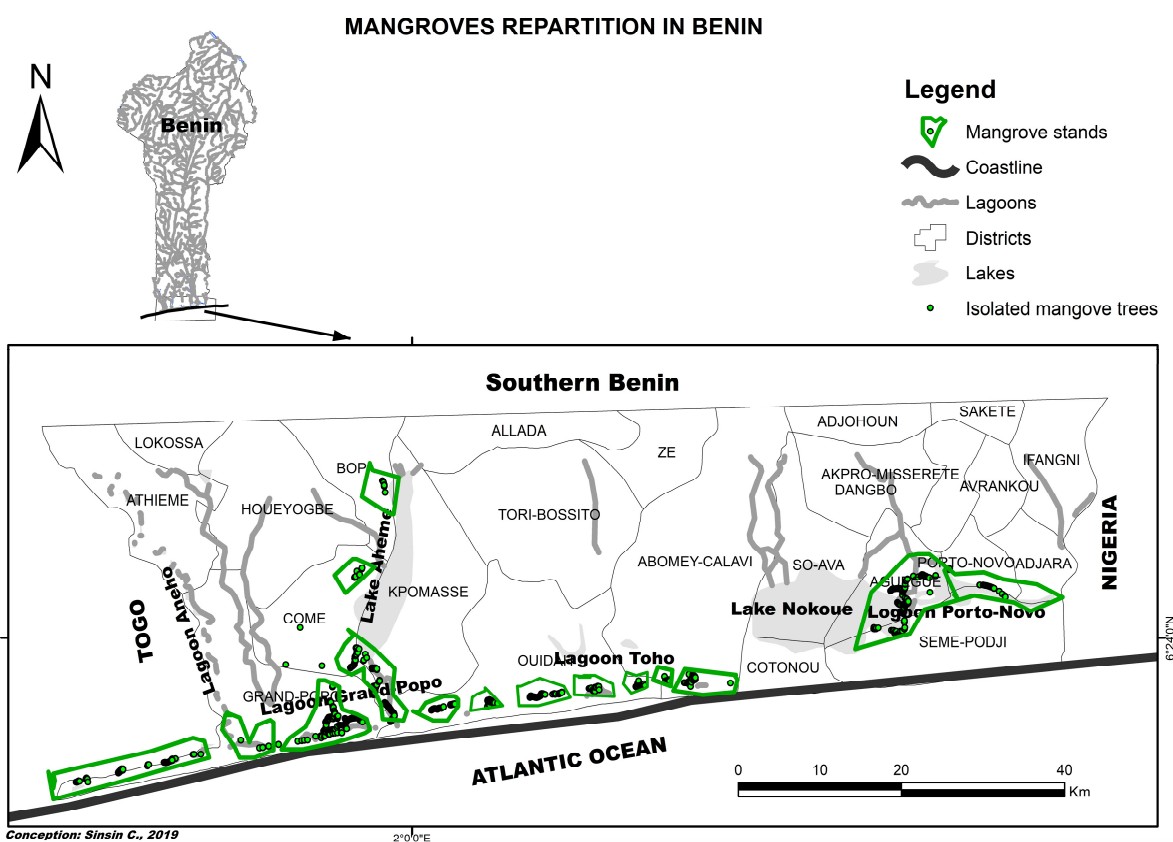

**Figure 1.** Map of the ecological niche of mangroves in Benin. **Source: Sinsin, C.B.L., 2019**.

### 2.2. Sampling and Data Collection

Fifteen villages were selected along the coastal region, with one village per each of the 15 subdistricts in which mangroves are found in Benin. A pre-survey on 30 interviewees, chosen at the household level, was conducted in each community to compute the number of interviewees to consider for the detailed survey. A structured interview was conducted using a prospective questionnaire with each selected informant. The interviewees were asked if they knew of any ecosystem service and the economic benefit of mangrove resources. The number of positive answers was 15. The proportion p of positive responses was considered for computing the sample size (*n*) as indicated in the formula of Equation (1) [26]

$$n = \frac{U_{1-\frac{\alpha}{2}}^2 \ x \ P(1-P)}{d^2} \tag{1}$$

where:

$U_{\frac{1-\alpha}{2}}$ is the normal approximation of the binomial distribution ($\alpha = 0.05$; $U_{\frac{1-\alpha}{2}} = 1.96$); *d* is the margin of error for any parameter to calculate for the study, fixed at 0.06; and *P* is the likelihood of those who knew of any ESs or the economic potentiality of mangroves ($P = 0.5$ = number of yes/total number of respondents for the exploration).

By replacing these figures in the formula of Equation (1), the sample size was then obtained as

$$n = \frac{1.96^2_{x\ 0.5\ x\ 0.5}}{0.06^2} = 267$$

The sample size (*n*) was found to be equal to 267, which was rounded up to 300 households. However, two persons did not provide a complete answer to the questions; hence the final sample size was 298. The 298 participants were equally distributed within the 15 villages, with 19–20 households per village. In addition, heads of NGOs active in the conservation and management of mangroves in Benin (e.g., Eco Benin, Nature Tropicale, Bees, and ONG-CORDES) were interviewed. National researchers that have demonstrated interest in mangrove ecosystems were identified and interviewed. As the number of NGOs and researchers involved in mangrove-related activities is low, they were all systematically considered in the study. Moreover, one local market was randomly selected per sub-district where mangroves are found (15 in total), and four mangroves' derivative product sellers (a total of 60 for the 15 local markets) were interviewed per market regarding the price of goods/substitutes.

First, data on ESs from mangroves were collected using a questionnaire and a meta-regression analysis approach. In each of the 15 villages, a short questionnaire (see Supplementary Materials) was administered to the 19–20 household heads (selected using a random selection procedure) in the presence of other household members. Answers were then perused and used as a baseline to build a questionnaire for the economic valuation. As for the economic valuation, questionnaires were designed separately according to the category of stakeholders. Community members, NGOs, researchers, and traders were interviewed (see Supplementary Materials). The total economic valuation approach [14] was adapted to recommendations from [22] to design questions for each ES and stakeholder category (Table 1). Market price, travel cost, contingent valuation, opportunity cost, damage-avoid cost, benefit transfer, production function, and meta-regression were thus used to obtain estimates of the economic value of the ESs. Where the contingent valuation method was applied, respondents were asked open/closed "willingness to pay" questions.

**Table 1.** Economic valuation of mangrove ESs: methods and techniques.

| Nature of the Service | Ecosystem Services | Category of the Service | Valuation Approach | Valuation Method | Description of the Method | Source |
|---|---|---|---|---|---|---|
| Direct use | Firewood | Provisioning | Market valuation | Market price | The market price of the good/substitute | [18,22,23,27] |
| | Timber | | | | | |
| | Branches for "Acadja" | | | | | |
| | Fodder | | | | | |
| | Handicraft (dyes to colour nets) | | | | | |
| | Fisheries | | | | | |
| | Crab collection | | | | | |
| | Oyster collection | | | | | |
| | Shrimp collection | | | | | |
| | Hunting (snakes, varan, etc.) | | | | | |
| | Ecotourism | Cultural and amenity services | Revealed preference | Travel cost | Direct and opportunity costs of time of visitors | [18,21] |
| | Worship | | | Contingent valuation | Willingness to pay | [12,28] |
| | Education | | | | Values from the literature and case studies in Africa | [12,22] |
| | Research | | | Meta-regression analysis | Values from African case studies and research interviews (Laboratory of Applied Ecology, Inspection forestière, etc.) | [12,28] |

**Table 1.** *Cont.*

| Nature of the Service | Ecosystem Services | Category of the Service | Valuation Approach | Valuation Method | Description of the Method | Source |
|---|---|---|---|---|---|---|
| Indirect Use | Carbon sequestration | Regulation | Market valuation | Market price | Price of the service | [10,12] |
| | Air purification | | | Opportunity cost | Costs that could be used to pay for other purification techniques | [12,22,28] |
| | Water purification | | | Opportunity cost | Costs that could be used to pay for other purification techniques | [8,12,21,28] |
| | Temperature regulation | | | Opportunity cost | Cost of other micro-climate cooling systems | [8,12,28] |
| | Waste treatment | | | Opportunity cost | Costs that could be used to pay for other water cleaning techniques | [8,12,28] |
| | Shoreline protection (flood | | | Opportunity cost and damage-avoid cost method | Costs that are avoided through their existence: e.g., wall construction costs and repairing damage that floods could cause to households if mangroves were not present | [8] |
| | Pollination | Habitat | Market valuation | Market price | Contribution of the ES to the delivery of other marketable goods/service | [12,22] |
| | Apiculture | | | Market price, production function | Contribution of the ES to the delivery of other marketable goods/service | [12,22] |
| | Aquaculture | | | Market price, production function | Contribution of the ES to the delivery of other marketable goods/service | [12,22] |
| | Nursery ground for fish | | | Production function | Contribution of the ES to the delivery of other marketable goods/service | [12,22] |
| Non-use | Biodiversity host | | Stated preference | Contingent valuation | Willingness to pay | [12,28] |
| | Existence | Cultural and amenity services | Value transfer | Benefit transfer | Transfer of benefits from a policy site to the study site | [12,28] |
| | Bequest | | | | | [23,28] |

*2.3. Data Analysis*

2.3.1. Economic Value Estimation

After tabulating the inventory questionnaires, a complete list of ESs from mangroves was obtained and grouped into categories as in [22]. The function from which the service/good derives and the nature of the service were the main criteria used for categorisation. Thus, ten categories of services were considered: group 1—existence, presence, and biodiversity of host; group 2—fisheries: fish, water crab, oyster, and shrimp; group 3—forestry: firewood, timber, and "acadja"; group 4—shoreline protection; group 5—carbon sequestration; group 6—ecotourism, research, and education; group 7—temperature regulation, water purification, waste treatment, and air purification; group 8—craft, folders, and medicine; group 9—nursery ground, pollination, hunting, bush crab, apiculture, and aquaculture; and group 10—worship.

Data collected from communities were used to compute economic values of services from groups 1, 2, 3, 7, 8, 9, and 10. For these categories of ESs, economic values were computed per group as follows:

- Household economic value per group of ESs (HEV$_G$)

The household economic value was computed for each group of services using Equation (2).

$$HEV_{G\alpha} = \sum_{i=1}^{n} EV_{\alpha i} \qquad (2)$$

where $HEV_{G\alpha}$ (USD) is the household economic value for the group (G) $\alpha$ (in the context of this study, $\alpha$ varies from 1 to 10); $EV_{\alpha i}$ (in USD) is the economic value of the service $i$ from

the group $\alpha$; and *i* represents an ecosystem service. For each group, *i* varies from 1 to the number of ESs ranged in the group $\alpha$.

- Economic value per group of ESs

The economic value is computed for each district ($EV_{G\alpha\beta}$) using Equation (3); and the economic value for each group ($EV_{G\alpha}$) is obtained through Equation (4).

$$EV_{G\alpha\beta} = \frac{1}{N}\sum_{i=1}^{n} \frac{HEV_{G\alpha}}{n_i \times S_\beta} \tag{3}$$

$$EV_{G\alpha} = \frac{\sum_{i=1}^{z} EV_{G\alpha\beta}}{z} \tag{4}$$

with *n* the number of households considered in the district $\beta$; $n_i$ the size of the household *i*; *S* the mangroves' surface coverage of the district $\beta$; and *z* the number of villages considered for the interviews.

For services in group 6, the information was derived from questionnaires administered to researchers, NGOs, and heads of schools. Its economic value was computed using Equation (5).

$$EV_{G6} = \frac{1}{n_i}\sum REV_i + \frac{1}{n_j}\sum EEV_j + \frac{1}{n_i + n_j}\sum LEV_k \tag{5}$$

$EV_{G6}$ (in USD) is the economic value of services from group 6; $n_i$ is the number of researchers' respondents; $REV_i$ (in USD) is the economic research value for the respondent *i*; $n_j$ is the number of NGO respondents; $EEV_i$ (in USD) is the ecotourism economic value for the respondent *j*; and $LEV_k$ (in USD) is the education's economic value for the respondent *k* (*k* is either a respondent *i* or a respondent *j* since respondents from both categories were considered for the ES "education").

Carbon storage (group 5) estimations were extracted from the regional blue carbon (BC) scheme [29]. Extracted data were extrapolated to the scale of Benin using national mangrove coverage from [25] and the REDD+ average carbon market price of USD 4.20 [30]. Thus, the economic value of group 5 ($EV_{G5}$ in USD) was computed using Equation (6).

$$EV_{G5} = CS \times PriceC \tag{6}$$

With *CS*, the carbon storage of mangroves (in metric tons) and *PriceC* (in USD/metric tons) is the average market price of BC.

Restrictive values of shoreline protection (group 4) were deduced from [31].

Afterwards, the total economic value (*TEV* in USD) was calculated using Equation (7).

$$TEV = \sum_{i=1}^{n} EV_{Gi} \tag{7}$$

where $EV_{Gi}$ (in USD) is the economic value of ESs from group *i* and *n* is the overall number of groups of ESs.

### 2.3.2. Statistical Analyses

The contribution (in per cent) of each group of services to the overall economic value of mangroves was calculated and plotted. A Principal Component Analysis (PCA) was applied to the contribution data to describe the relationship between groups of ESs. This analysis allows us to determine convergent (synergy) ESs and non-convergent ESs (trade-off). Such research could advise the group of ESs of priority if one wishes to optimise economic gain from mangroves while applying strong conservation measures for sustainable management. The analysis was conducted with the package "FactoMineR" [32].

A Linear Model with Mixed Effects (LMME; Equation (8)) was used to identify which socio-demographic factors determine the economic value of mangroves. "Village" was considered a grouping factor, and the model was implemented with the function lmer () with the package "leme4" [33].

All statistical analyses were carried out in R 4.1.2 [34].

## 3. Results

### 3.1. Ecosystem Services (ESs) from Mangroves in Benin

Twenty-nine (29) Ecosystem Services (ESs) were cited by interviewees during the first round of interviews. These are in the following order: existence (93.6%), presence (92%), fish (89.3%), water crabs (83.2%), firewood (79.9%), biodiversity host (79.9%), temperature regulation (76.2%), nursery ground (72.8%), shrimp (72.5%), air purification (72.5%), ecotourism (72.2%), wood manufacturing (71.5%), "acadja" (69.8%), pollination (69.2%), bush crabs (68.8%), research (67.1%), education (52.7%), water purification (44.3%), shoreline protection (44%), crafting (39.3%), oysters (35.6%), aquaculture (33.9%), folders (30.2%), hunting (19.1%), honey (18.5%), worship (15.8%), carbon storage (15.1%), waste treatment (15.1%), and medicine (1.3%). Identified ESs were categorised into ten groups as described in Section 2.1. (See Table 2).

**Table 2.** Ecosystem Services from Benin mangroves grouped by categorisation adopted from [22].

| Group | ESs | Citation Rate (%) | Citation Rank |
|---|---|---|---|
| 1 | Existence | 93.6 | 1 |
|  | Presence | 92 | 2 |
|  | Biodiversity host | 79.9% | 6 |
| 2 | Fish | 89.3 | 3 |
|  | Water crabs | 83.2 | 4 |
|  | Shrimp | 72.5 | 9 |
|  | Oysters | 35.6 | 21 |
| 3 | Firewood | 79.9 | 5 |
|  | Wood manufacturing | 71.5 | 12 |
|  | "Acadja" | 69.8 | 13 |
| 4 | Shoreline protection | 44 | 19 |
| 5 | Carbon sequestration | 15.1 | 27 |
| 6 | Ecotourism | 72.2 | 11 |
|  | Research | 67.1 | 16 |
|  | Education | 52.7 | 17 |
| 7 | Temperature regulation | 76.2 | 7 |
|  | Water purification | 44.3 | 18 |
|  | Waste treatment | 15.1 | 28 |
|  | Air purification | 72.5 | 10 |
| 8 | Craft | 39.3 | 20 |
|  | Folders | 30.2 | 23 |
|  | Medicine | 1.3 | 29 |
| 9 | Nursery ground | 72.8 | 8 |
|  | Pollination | 69.2 | 14 |
|  | Hunting | 19.1 | 24 |
|  | Bush crabs | 68.8 | 15 |
|  | Apiculture | 18.5 | 25 |
|  | Aquaculture | 33.9 | 22 |
| 10 | Worship | 15.8 | 26 |

*3.2. The Monetary Value of Each Category of Service and the Global Annual Wealth*

In Benin, mangroves provide multiple services to human communities in their surroundings. ESs of group 7 (USD 115,284.9/hectare) followed by those of group 6 (USD 59,387.7/hectare) have the highest economic values, while ESs of group 3 (USD 0.0/hectare) were proved to have little economic importance (Table 3). The total annual monetary value of mangrove ecosystems in Benin was estimated at USD 1.29 billion, based on a rate of USD 195,223.69/hectare.

**Table 3.** Estimated economic value of mangrove ecosystem services (USD/hectare).

| Groups | Abomey-Calavi | Aguégués | Sèmé-Kpodji | Porto-Novo | Comé | Bopa | Grand Popo | Ouidah | Kpomassè | Sô Awa | Mean EV |
|---|---|---|---|---|---|---|---|---|---|---|---|
| Group 1 | 1148 | 98 | 113 | 310 | 174 | 7 | 3 | 243 | 122 | 82 | 230.0 |
| Group 2 | 21330 | 1451 | 6012 | 7347 | 8778 | 1170 | 679 | 10,807 | 3235 | 7016 | 6782.5 |
| Group 3 | 0 | 0 | 0 | 0 | 0 | 0 | 0 | 0 | 0 | 0 | 0 |
| Group 4 | 3217 | 9650 | 3217 | 3217 | 4825 | 6433 | 643 | 1930 | 4825 | 4825 | 7173.2 |
| Group 5 | 1820 | 1820 | 1820 | 1820 | 1820 | 1820 | 1820 | 1820 | 1820 | 1820 | 1820 |
| Group 6 | 112,289 | 85,174 | 95,733 | 90,454 | 47,299 | 24,125 | 18,364 | 49,156 | 18,913 | 52,370 | 59,387.7 |
| Group 7 | 230,828 | 31,482 | 39,056 | 152,138 | 31,624 | 764 | 100 | 620,723 | 12,850 | 33,284 | 115,284.9 |
| Group 8 | 0 | 0 | 0 | 0 | 0 | 0 | 0 | 132 | 0 | 0 | 13.2 |
| Group 9 | 6333 | 178 | 33,571 | 92 | 1059 | 196 | 112 | 883 | 394 | 1733 | 4455.1 |
| Group 10 | 0 | 0 | 30 | 413 | 254 | 0.07 | 0.78 | 24 | 29 | 20 | 77.09 |
| Total | | | | | | | | | | | 195,223.69 |

Source: f = Field data.

ESs contribute at different rates to the total economic value of mangroves (Figure 2). ESs of group 7 (temperature regulation, water purification, waste treatment, and air purification: ~59%) followed by those of group 6 (ecotourism, research, and education: ~29%) had high economic values while those of group 3 (firewood, timber, and "acadja": 0%) were reported having nominal economic value. In addition, ESs from groups 2, 4, 5, and 9 (fisheries, shoreline protection, carbon sequestration, nursery ground, pollination, hunting, bush crabs, apiculture, and aquaculture) revealed relative weak economic importance. Very few respondents acknowledged the economic significance of those of groups 1, 8, and 10 (existence, presence, biodiversity host, craft, folder, medicine, and worship).

Furthermore, the districts' contribution to mangrove ecosystems' total economic value (TEV) is uneven (Figure 2). Overall, the communities of Ouidah (34.36%) and Grand Popo (1.09%) contribute at the highest and lowest rates to the TEV.

The Principal Component Analysis (PCA) results show that the TEV of mangroves is controlled by two principal components (PC or Dim) of services: Dim1 (33.1%) and Dim2 (21.9%) (Figure 3). Given that ESs of group 10 (worship services) and group 4 (shoreline protection services) have the best representation in Dim1, it could be assumed that Dim1 is representative of ESs with no direct market price and no direct income to local communities. Yet, ESs with direct market price and direct income benefits to stakeholders (groups 1, 2, 7, and 6) are relatively well represented in the second principal component (Dim2). ESs of groups 9, 10, and 5 are less represented and contribute less to the TEV (lowest values of cos2) while those of groups 8, 7, and 4 show average contribution to the TEV. However, high values of cos2 for groups 1, 2, and 6 show that ESs of these groups represent a significant part of the overall economic value of mangroves in Benin (Figure 3). Thus, ecosystem services such as existence, presence, biodiversity host, fisheries (fish, water crabs, shrimp, and oysters), ecotourism, research, and education are relevant to the majority of stakeholders through either direct or indirect uses. Moreover, the high positive correlation between these ESs (groups 1, 2, and 6) suggests that the valuation of either of these does not hamper the sustainability of others. Similarly, services of group 7 show a high correlation with those of group 8. Regarding Dim1, the benefits of groups 1, 2, 6, 7, and 8 can be given priority together without compromises. However, whichever valuation

aspect is considered, the services of group 4 correlate less with those of groups 1, 2, 7, and 8. Nonetheless, regarding dimension 2, the services of groups 4 and 6 appear to be dependent.

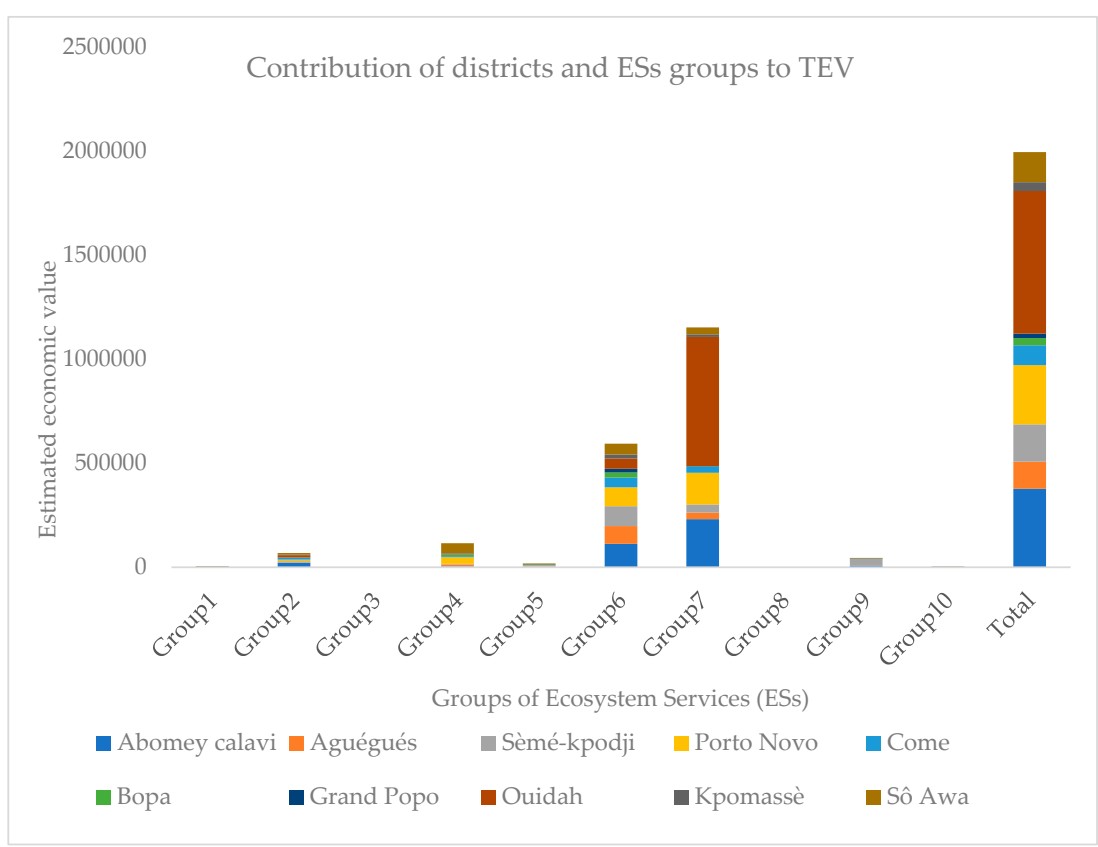

**Figure 2.** Contribution of ESs to the total economic value of mangroves.

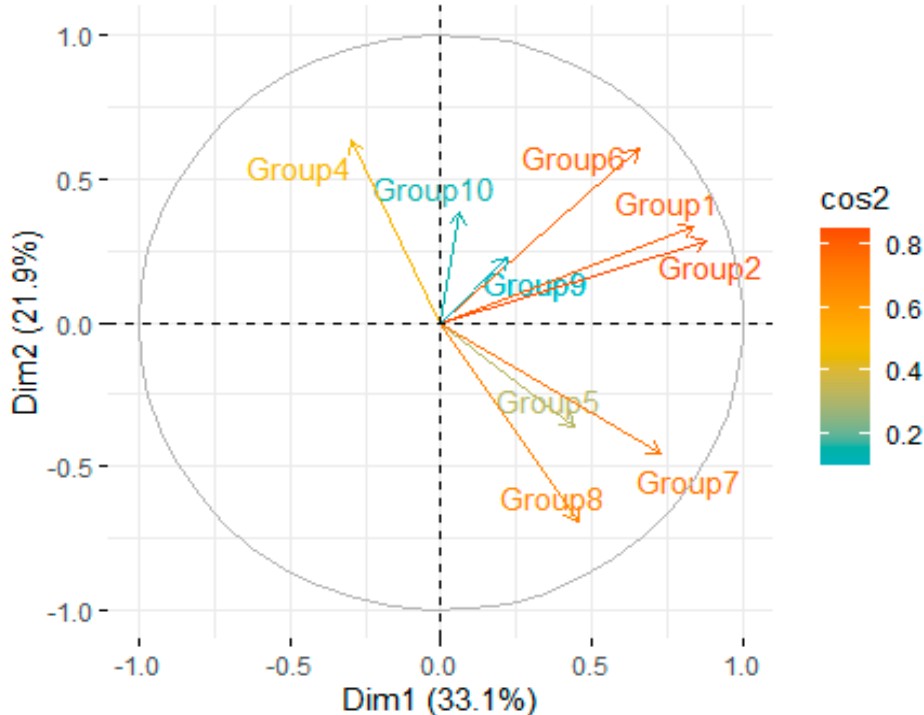

**Figure 3.** Results of the Principal Component Analysis showing correlations between groups of ESs.

### 3.3. Drivers of Mangroves' Economic Value

The Linear Mixed Effect Model (LMEM) indicates that only management stakeholders followed by household size are meaningful predictors of the TEV (Table 4). Estimates of the model (Table 5) show that management stakeholders and household size negatively correlate to the TEV. This trend means that the less a respondent household is implicated in mangroves' management activities, the higher the income from mangroves' ecosystem services; and the smaller the household size is, the higher the household's income from mangroves.

$$lmer(formula = Economic\ value \sim sex + age + years\ of\ residence +$$
$$marital\ status + ethnicity + religion + education + household\ size +$$
$$main\ activity + years\ in\ fishery + fishery\ type + secondary\ activity +$$
$$income\ source + appurtenance\ to\ management\ structure +$$
$$conservation\ training + public\ awareness + private\ awareness +$$
$$(1|Village)) \tag{8}$$

**Table 4.** Results of the ANOVA for the Linear Mixed Effect Model.

|  | df | Chi$^2$ | *p* Value |
|---|---|---|---|
| Sex | 1 | 1.8282 | 0.176340 |
| Age | 1 | 0.4892 | 0.484271 |
| Years of residence | 1 | 0.1536 | 0.695165 |
| Marital status | 4 | 0.9542 | 0.916655 |
| Ethnicity | 8 | 0.6712 | 0.999595 |
| Religion | 4 | 5.0551 | 0.281694 |
| Education | 6 | 0.9136 | 0.988676 |
| Household size | 1 | 3.7667 | 0.052284. |
| Main activity | 2 | 1.0343 | 0.596205 |
| Years of fishery | 1 | 1.9731 | 0.160117 |
| Fishery type | 3 | 4.6484 | 0.199433 |
| Secondary activity | 4 | 0.7695 | 0.942491 |
| Income source | 6 | 9.6597 | 0.139735 |
| Management stakeholder | 3 | 12.7043 | 0.005322 ** |
| Conservation training | 3 | 0.0578 | 0.996366 |
| Public awareness | 1 | 0.0144 | 0.904362 |
| Private awareness | 1 | 0.0928 | 0.760656 |
| RMSE |  | 3891.228 |  |

** Indicates significance at 5%.

**Table 5.** Summary of the Linear Mixed Effect Model.

|  | Estimate | Standard Error | T Value |
|---|---|---|---|
| Intercept | 3738.78 | 7152.75 | 0.52 |
| Sex | 2607.35 | 1928.35 | 1.35 |
| Age | −20.21 | 28.89 | −0.70 |
| Years of residence | 8.11 | 20.7 | 0.39 |
| Marital status | −4328.68 | 4909.86 | −0.88 |
| Ethnicity | 2136.58 | 5234.09 | 0.41 |
| Religion | 2749.38 | 3659.17 | −0.64 |

**Table 5.** *Cont.*

|  | Estimate | Standard Error | T Value |
| --- | --- | --- | --- |
| Education | 507.90 | 3178.99 | 0.16 |
| Household size | −128.759 | 66.34 | −1.94 |
| Main activity | 884.56 | 3447.00 | 0.26 |
| Years of fishery | 42.41 | 30.19 | 1.40 |
| Fishery type | 1595.58 | 1030.55 | 1.55 |
| Secondary activity | 1874.20 | 3287.88 | 0.57 |
| Income source | 939.155 | 1404.726 | 0.669 |
| Management stakeholder | −4730.83 | 2573.98 | −1.84 |
| Conservation training | 436.12 | 3562.89 | 0.12 |
| Public awareness | −30.30 | 252.196 | −0.12 |
| Private awareness | −23.24 | 76.31 | −0.31 |

## 4. Discussion

Up to 29 ESs corresponding to provisioning and regulation functions were extracted from mangrove ecosystems in Benin. This finding matches previous global descriptions of mangroves' ESs [12,23,35] and the diversity of ESs in tropical mangroves [36]. Nonetheless, the findings of this study show a slight difference in the scale of goods' provisions. For instance, bush crab species were reported as fishery resources, and our findings confirmed results of a recent study on the district of Grand Popo in Benin [37]. Unlike upland ecosystems, no plant part was reported directly edible, which might discredit the "willingness to pay" value granted to this system despite the few medicinal plants they provide [38,39]. Great attachment to bequest value that has no direct monetary reward to local communities could possibly explain the priceless importance of mangrove ecosystems by indirect outcomes such as opportunities for the pharmaceutical industry [40], tourism [11,41], carbon price [42], etc. However, to fill the gap in quantifying mangroves' value to inform and convince decision-makers, it is worth considering the detailed goods and services attached to their existence and presence.

Fine-scale valuation of ESs highlighted regulation (water, air, waste, and temperature) followed by recreation services as the most valuable. This trend challenges the idea that recreation services are often ranked at the top [22]. Indeed, enforcement of national conservation policies (Ministerial council at its session of 26 October 2016) prohibiting all activities in mangroves could explain the non-attribution of economic value to forestry services (wood and timber extraction) and the non-contribution of services of group 3 to the TEV (Figure 2). The success of this decree would progressively remove forestry ESs from Benin mangroves, which may be a good step towards sustainability. In contrast, fishery income is relatively high compared with that found in investigations from Kenya [8] and Nigeria [6], both tropical mangroves. The minor importance given to mangroves as worship patrimony (only 15% of respondents and USD 77/hectare/year) confirms the very low representability of their economic value in the mangrove valuation literature [23]. This could be explained by limitations in the methodological approach. Despite the low monetary value of carbon storage service (USD 1820/hectare/year; less than 1% contribution to TEV), it is worth highlighting and communicating this to climate negotiation institutions while waiting for further detailed studies. To that end, policy briefs and workshops could be helpful.

A good understanding of the complex interrelationships between social and natural systems and the multiple dimensions and different time scales of ecosystem services is crucial for optimising benefits from mangrove ecosystems while ensuring effective conservation [43]. Results of the PCA could be used to define priority lines for ES valuation. Indeed, projection of shoreline protection (group 4) on either axis suggests that optimising economic income from this service may reduce economic benefits from others; mainly

regulation (group 7), ethnobiology (group 8), recreation (group 6), provision (group 2), and existence (group 1) services. Undeniably, priority to shoreline protection may induce less anthropogenic activity within mangroves and, by extension, this could reduce other goods and services benefits to human communities. Therefore, to meet the aspirations of the United Nations Sustainable Development Goals agenda [44], there is a need for a trade-off between services to promote conservation while enhancing human well-being and livelihoods.

The relatively high annual monetary value of Benin mangroves (USD 195,223.69/hectare for only 0.04% of the world's mangrove coverage) compared with global trends (USD 2000 to USD 200,000/hectare [25]) and those of other countries (USD 2936/hectare for Fiji [45]; USD 789.5/hectare for Mexico [46]; USD 4443.5/hectare for Malaysia [28]; USD 71.5/hectare for Thailand [47]; USD 1287/hectare for Pakistan [48]; USD 2212/hectare for Southeast Asia [12]) could be an indicator that coastal communities of Benin have a great connection to mangroves. Therefore, mangroves could serve as a potential path for the coastal economic growth of Benin in balance with other countries. The overall annual value (USD 1.29 billion for 6600 hectares) is higher than in countries with massive mangrove coverage–e.g., Vietnam, with USD 301–503 million/year for 157,500 ha [14]—suggests that Benin mangroves are highly productive and play paramount roles for coastal communities' well-being. This recalls the need to reinforce conservation measures and increase restoration efforts to reduce wealth loss and improve local and national economic figures.

Not all socio-economic factors considered in a study effectively impact the outcomes. Some socio-economic factors are determinants for understanding the explained variable, while some have no significant effects. For instance, a Multiple Linear regression model proved that age, marital status, household size, education, and period of residence determined exploitation patterns of mangrove resources in Zanzibar. At the same time, gender and income were revealed not to influence the exploitation of these resources [49]. For our study, use of a Linear Mixed Effect Model showed that appurtenance to management structures followed by household size are primary drivers of the total economic value. Hence, demographic factors (household size) and management (appurtenance to management structures) should be referred to for the procurement of human resources from local communities if the government ever decides to manage mangrove ecosystems.

In addition, the vast difference in value estimation could be due to the methodological approach. Most valuation studies used benefit transfer/meta-regression, which often results in an overestimation/underestimation of price [18]. This study has the merit of using field data collection to reflect the actual contribution to the national economy and household income. It has also addressed concerns of [23] regarding the accuracy of valuation methods in providing motivating conservation tools to stakeholders, especially to decision-makers.

## 5. Conclusions

This study has the merit of providing reliable information on Ecosystem Services (ESs) from mangroves in Benin and on their economic value. Overall, Benin mangroves are important wealth sources that provide up to 29 ESs whose economic value is estimated at USD 195,223.69/hectare/year for a total of USD 1.29 billion/year. These ESs are of high economic importance to communities in Ouidah and Grand-Popo. The economic value of mangroves is driven by geographic position, human demography, cultural background, and stakeholdership. The wide range of ESs provided by mangroves in Benin ranks them as important resources in both the mitigation and the adaptation to climate change strategies. Results of this study are thus relevant for all stakeholders, including national and international institutions investing in climate protection and climate change adaptation, NGOs dedicated to mangroves conservation, local communities living in the mangrove areas, and researchers. It is recommended that further studies be conducted on the carbon sequestration economic value using real-time field data to capitalize on the mangroves in Benin in carbon market deals.

**Supplementary Materials:** The following are available online at https://www.mdpi.com/article/10.3390/land12051094/s1.

**Author Contributions:** Conceptualization, C.B.L.S.; methodology, C.B.L.S. and A.B.; software R.L.G.K.; validation, K.V.S. and R.L.G.K.; formal analysis, C.B.L.S.; investigation, C.B.L.S.; resources, C.B.L.S.; data curation, C.B.L.S.; writing—C.B.L.S., A.B., K.V.S., R.C.G. and R.L.G.K.; visualization, all authors; supervision, R.L.G.K.; project administration, C.B.L.S.; funding acquisition, C.B.L.S. All authors have read and agreed to the published version of the manuscript.

**Funding:** Data collection of this work was conducted with the support of the International Foundation for Science (IFS), under the grant number D_6309. This work was published with the aid of a grant in the UNESCO-TWAS programme, "Seed Grant NO. 4500474950 for African Principal Investigators" financed by the German Federal Ministry of Education and Research (BMBF). The views expressed herein do not necessarily represent those of UNESCO-TWAS nor BMBF.

**Data Availability Statement:** The data presented in this study are available on request from the authors. Data were collected by the authors.

**Acknowledgments:** Authors are grateful to Montcho Yvette, Agonvonon Plottin, and Sinsin Renaud who assisted in the data collection.

**Conflicts of Interest:** The authors declare no conflict of interest. The funders had no role in the design of the study; in the collection, analyses, or interpretation of data; in the writing of the manuscript; or in the decision to publish the results.

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
