# Peer review of "Economic Valuation of Mangroves and a Linear Mixed Model-Assisted Framework for Identifying Its Main Drivers: A Case Study in Benin"

_land, doi:10.3390/land12051094_

Round 1

Reviewer 1 Report

Comments on the attached file

English can be improved

Reviewer 2 Report

The manuscript “Economic valuation of mangroves and a linear mixed model-assisted framework for identifying its main drivers: a case study in Benin” presents a study about the always interesting economic valuation of ecosystem services, in this case, applied to mangroves in Benin.

First of all, I would like to congratulate the authors for their work. I am aware of the difficulties that arise in these works and yet of their importance in raising social awareness of the relevance of ecosystems.

In spite of this, I consider that the manuscript can be improved in some points, some critical, before being published in the journal. To structure my review, I will be guided by and follow the questions proposed in the guidelines for reviewing.

Is the manuscript clear, relevant for the field and presented in a well-structured manner?

Yes, there are no concerns about these points in the present form of the manuscript. The structure is correct in my point of view.

Are the cited references mostly recent publications (within the last 5 years) and relevant? Does it include an excessive number of self-citations?

Everything is correct regarding these points. Regardless of the relative age of the articles cited, reference has been made to articles of relevance in the field of valuation of ecosystem services, especially those applied to mangroves. Global, continental, and national references are included.

Is the manuscript scientifically sound and is the experimental design appropriate to test the hypothesis?

Are the manuscript’s results reproducible based on the details given in the methods section?

I am going to answer these two questions together, as these could be some of the critical points of my review.

In my humble opinion, I think the methodology is at some point, confusing. If I understand correctly, for most of the groups of ecosystem services identified, the valuation is obtained through surveys of the agents/stakeholders involved and the affected population. Does this mean that a contingent valuation has been used? If not, what approach has been followed?

At this point, I miss some more information on this point, specifically on the questions that have been asked for the identification and valuation of the services provided by mangroves.

Were open-ended, multiple-choice, and open-ended questions used, and were respondents provided with a range of values?

The following reference provides much information on how to obtain and conduct ecosystem valuation surveys (Pearce, David, Atkinson, Giles and Mourato, Susana (2006) Cost-benefit analysis and the environment: recent developments. Organisation for Economic Co-operation and Development, Paris, France. ISBN 9264010041)

On the other hand, Table 1 identifies multiple ecosystem services, along with valuation methods and references. Does this mean that a benefit transfer has been used?

As I say, I am not at all clear about the methodology followed at this point for obtaining the valuations and what use is made of the surveys. I suggest much more clarification on this point in the manuscript and perhaps a brief explanatory diagram might help to better understand the methodology followed.

Are the figures/tables/images/schemes appropriate? Do they properly show the data? Are they easy to interpret and understand? Is the data interpreted appropriately and consistently throughout the manuscript? Please include details regarding the statistical analysis or data acquired from specific databases.

Some suggestions regarding figures:

1. Figure 1: maybe the location of mangroves could be shown on the map.

2. In section “Economic Valuation Estimation”, I suggest describing the categories/groups by using a table. The same information is repeated in Section 3, so consider omitting the repetition.

3. Figure 3: What are the cos2, Dim1 and Dim2 represented?

Are the conclusions consistent with the evidence and arguments presented?

I believe that the conclusions should be developed further. In the current version of the manuscript they merely repeat the assessment obtained and make a recommendation to the Ministry of Agriculture of Benin (in my opinion, very direct and perhaps unusual in research work).

I suggest that the authors focus on describing the degree to which the objectives of the study were achieved, highlighting the main results, and above all, on explaining the relevance and usefulness of the study. Why is the finding important? What is it useful for? Who is it interesting for and why? What could be done on the basis of the results obtained?

Apart from the above, I would like to add some additional comments that may help to improve the quality of the study:

- Revise the numbering of the headings. It is not correct. Neither is the numbering of the lines of the manuscript (which at some point is reset), although it is a minor point.

- The section "Sampling and Data collection", talks about "the 15 villages". Which villages are these? Where are they located? Why and how have they been selected?

- The sentence "The total economic valuation approach [14]..." is not clear enough. What approach is followed here?

- In the Section "Economic Value Estimation", in Equation 5, REV is "The economic research value for the respondent". Does this mean that a contingent valuation is being used? As I noted above, the methodology followed needs to be better explained.

- I suggest incorporating an example of the questionnaire used.

- In the analysis of the results, it might be interesting to add a comparison between the presence of ecosystem services, the importance for the agents/stakeholders/population, and the final valuation of each of them.

Some minor typos and a rephrasing review.

Round 2

Reviewer 2 Report

Thanks to the authors for the revised version of their manuscript, and thanks for the work done, taking into consideration my comments and those of other reviewers.

I consider that the manuscript has increased in quality, and therefore I recommend that it be published in the journal.

However, there are some things that could still be improved and that would help to have a higher quality and more interesting paper for readers and other researchers.

1.- I believe that the approach followed with the surveys and the use of contingent valuation is not sufficiently explained. In my opinion, this explanation should be given more weight and more detail.

2.- Perhaps due to lack of knowledge, but I find the meaning and interpretation of the PCA difficult to understand. What exactly is being represented? How should the results be interpreted?

3.- Perhaps the conclusions could be extended a little more.
